# Kilometer-scale structure on the core–mantle boundary near Hawaii

Zhi Li 🔗 [1 ✉], Kuangdai Leng[2,3], Jennifer Jenkins[1,4] & Sanne Cottaar[1]

The lowermost mantle right above the core-mantle boundary is highly heterogeneous containing multiple poorly understood seismic features. The smallest but most extreme heterogeneities yet observed are 'Ultra-Low Velocity Zones' (ULVZ). We exploit seismic shear waves that diffract along the core-mantle boundary to provide new insight into these enigmatic structures. We measure a rare core-diffracted signal refracted by a ULVZ at the base of the Hawaiian mantle plume at unprecedentedly high frequencies. This signal shows remarkably longer time delays at higher compared to lower frequencies, indicating a pronounced internal variability inside the ULVZ. Utilizing the latest computational advances in 3D waveform modeling, here we show that we are able to model this high-frequency signal and constrain high-resolution ULVZ structure on the scale of kilometers, for the first time. This new observation suggests a chemically distinct ULVZ with increasing iron content towards the core-mantle boundary, which has implications for Earth's early evolutionary history and core-mantle interaction.

[1] Bullard Laboratories, Department of Earth Sciences, University of Cambridge, CB3 0EZ Cambridge, UK. [2] Department of Earth Sciences, University of Oxford, OX1 3AN Oxford, UK. [3] Rutherford Appleton Laboratory, Science and Technology Facilities Council, Didcot OX11 0QX, UK. [4] Department of Earth Sciences, Durham University, DH1 3LE Durham, UK. ✉email: zl382@cam.ac.uk

The core–mantle boundary separates the Earth's liquid iron-nickel outer core from the solid silicate mantle. Heat from the core powers convection in the mantle, driving hot buoyant upwellings known as mantle plumes, that rise from the core-mantle boundary to the Earth's surface where they form hotspots and volcanoes. Over the past few decades, seismology has revealed that the mantle immediately above the core–mantle boundary is highly heterogeneous, potentially analogous to the level of variability seen at lithospheric and crustal levels near the Earth's surface[1].

Ultra-Low Velocity Zones (ULVZs) represent the most extreme core–mantle boundary features yet observed, generally showing shear-wave velocity reductions of 10 to 30%[2]. However, the small height of these structures - only tens of kilometers high - means they are below the resolution limit of tomographic models, requiring higher frequency seismic waves to image them. Currently available seismic data have limited earthquake-station geometries that can be used to image these structures and therefore only a fraction of the core–mantle boundary is illuminated. Despite this, a number of ULVZs varying in size, shape, and velocity reduction have been reported[2,3]. Early studies interpreted ULVZs as sporadic and localized structures, but found little constraint on their lateral extent[4–6]. More recently four unusually large ULVZs linked to areas of hotspot volcanism have been mapped near Hawaii[7,8], Iceland[9], Samoa[10], and the Marquesas[11]. Unlike earlier studies, which suggested ULVZs represented small-scale structures, these new observations indicate the presence of thin but wide ULVZs, on the order of 600–900 km across. These have been dubbed 'mega-ULVZs' in some literature[10,11]. The large aspect ratio of mega-ULVZs suggest they are very dense compared to their surroundings[12], which could be explained by iron enrichment[13,14]. Increasing evidence suggests that the correlation between the geographic location of mega-ULVZs and surface hotspots may not be a coincidence, i.e. the isotopic tungsten anomalies collected in various ocean island basalts indicate a primordial material or signatures from core infiltration in the mantle plume[15,16].

Three of the recently observed mega-ULVZs, beneath Hawaii, Iceland, and the Marquesas, have been discovered using core-diffracted shear waves known as $S_{diff}$[7,9,11]. $S_{diff}$ phases are observed at epicentral distances over 100°, and sometimes extending to up 140° after a long refraction path along the core–mantle boundary (Fig. 1A). $S_{diff}$ energy at higher frequencies and shorter wavelengths (on the order of the ULVZs height) which propagate closer to the core–mantle boundary, can get trapped in thin mega-ULVZs, becoming delayed and refracted. On a seismogram these guided waves appear tens of seconds after the main $S_{diff}$ phase, and for a cylindrical ULVZ show a hyperbolic delayed move-out as a function of azimuth[7]. We refer to these as $S_{diff}$ postcursors illustrated in Fig. 2. The travel-time delays of $S_{diff}$ postcursors hold information on the size, shape, and average velocity reduction of the ULVZ they sample[7]. Here we demonstrate that the frequency content and dispersive properties of these phases also contain information on the vertical internal velocity structure of ULVZs. Although $S_{diff}$ waves sample a reasonably large area of the core–mantle boundary, the observation of $S_{diff}$ postcursor waves with a hyperbolic move-out is still quite rare (though evidence of $S_{diff}$ postcursors without this characteristic, which may be caused by other lower mantle structures, have been observed across the Pacific[3,11]). This is partly due to the requirement of a continuous and dense array of seismic stations. The previous observations of mega-ULVZs can all be attributed to the deployment of the large-scale US Transportable Array[17].

In this work, we exploit a new compilation of postcursors for seismic shear waves that diffract along the core–mantle boundary

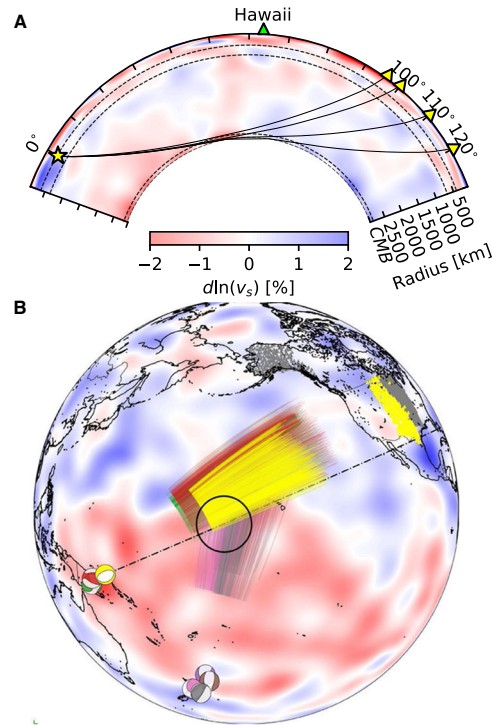

**Fig. 1 Events and Sdiff ray paths used in this study. A** Cross-section slicing the center of Hawaiian ultra-low velocity zone, showing ray paths of $S_{diff}$ waves at 96°, 100°, 110°, and 120° for 1D Earth model PREM[49]. The dashed lines from top to bottom mark the 410 km, 660 km discontinuity, and 2791 km (100 km above the core-mantle boundary). **B** Events and $S_{diff}$ ray paths on the background tomography model SEMUCB_WM1 at 2791 km depth[50]. Beachballs of events plotted in different colors including 20100320 (yellow), 20111214 (green), 20120417 (red), 20180910 (purple), 20180518 (brown), 20181030 (pink), 20161122 (gray), stations (triangles), and ray paths of $S_{diff}$ waves at pierce depth 2791 km in the lowermost mantle used in this study. The event used in short-period analysis is highlighted in yellow. Proposed ULVZ location is shown in black circle. Dashed line shows cross-section plotted in **A**.

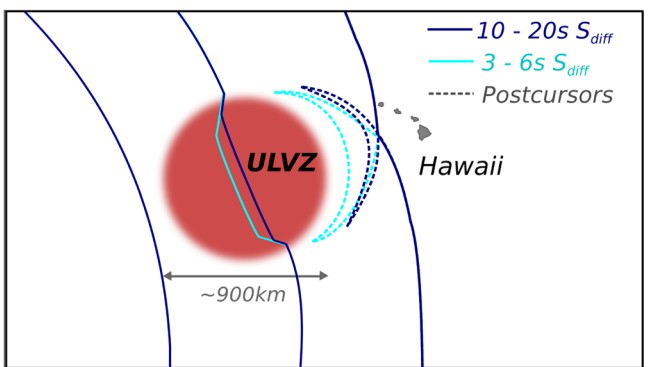

**Fig. 2 Illustration of wavefront refraction for the long-period (black) and short-period (green) Sdiff postcursors.** Sketch shows the wavefront propagating through a 900 km diameter cylindrical ultra-low velocity zone (ULVZ) (red) and caused delayed long-period (blue) and short-period (cyan) $S_{diff}$ postcursors.

that sample the Hawaiian ULVZ. We find strongly delayed postcursors at higher frequencies than previously observed. The postcursors have strong frequency-dependent sensitivity to structure at different length scales above the core–mantle boundary. Utilizing the latest computational advances in 3D

waveform modeling, we model this high-frequency signal and illuminate the internal ULVZ structure on the scale of kilometers at the core–mantle boundary.

## Results

**Location of the Hawaiian ULVZ**. When $S_{diff}$ postcursors illuminate a ULVZ from a specific direction, the exact location can be anywhere along the diffracted path, and there are trade-offs between location, size, and velocity reduction in the modeling of these structures. The Hawaiian ULVZ was initially identified using $S_{diff}$ waves from deep earthquakes in Papua New Guinea recorded at the transportable array and other stations in the central United States[7] (Fig. S1). This left uncertainty as to the exact location of the ULVZ in the NE-SW direction along the $S_{diff}$ path, and calls into question the reliability of the simplified cylindrical model proposed to fit the data. With the redeployment of the transportable array to Alaska starting in 2014, increased data coverage of $S_{diff}$ waves that highlight the Hawaiian ULVZ in the N-S direction is now available. Four deep earthquakes from the Kermadec trench recorded in Alaska show postcursor energy caused by the Hawaiian ULVZ (Fig S2). Using these new data, we are now able to pinpoint the precise location of the Hawaiian ULVZ (Fig. S3). The hyperbolic travel times of postcursors that characterize this dataset suggest that the ULVZ is centered at 172.3°W and 15.4°N—offset further southwest from the Hawaiian Islands than previously thought[7]. Although the exact shape of the ULVZ is not well-constrained given the current data coverage, synthetic modelings of the waveforms from two directions indicate a cylindrical shape is a good first approximation of the Hawaiian ULVZ (Figs. S1 and S2), similar to that proposed for the Icelandic ULVZ[9].

**High-frequency $S_{diff}$ postcursors**. Previously, $S_{diff}$ postcursors have been observed down to periods of 10 s (frequencies ≤ 0.1 Hz), which corresponds to a 500 km wide horizontal resolution (based on Fresnel zone half width) and limits the vertical resolution of imaged ULVZs to tens of kilometers. This is insufficient to unravel the details of internal ULVZ velocity gradients, which requires the exploitation of higher frequency observations. However, pushing $S_{diff}$ observations to higher frequencies is challenging: arrival amplitudes are weaker, background noise from ocean waves is louder (peaking at 0.14 Hz), and the computational costs for modeling full 3D synthetic data increases as power of frequency[18,19]. In this study, we make the first high-frequency observations of $S_{diff}$ postcursors down to 3 s (≤ 0.3 Hz), allowing us to infer internal ULVZ structure on the order of kilometers (Fig. S4).

We focus on the high-quality $S_{diff}$ data from a 2010 earthquake in the Papua New-Guinea area recorded in the contiguous United States (Fig. 1B in yellow). The $S_{diff}$ observations are filtered into two frequency bands for comparison, one from 10 to 20 s ('long-period'), and one from 3 to 6 s ('short-period'). The energy of the short-period phase is very weak; it is barely observable in the raw data (Fig. S5). To enhance the signal-to-noise ratio, we apply a subarray phase-weighted beamforming technique for each station and its nearest 20 neighbors, which stacks the signals based on phase and directional coherency ("Methods", linear stack result in Fig. S6). Figure 3 shows the results of this phase-weighted stacking in separate time windows for the main $S_{diff}$ arrivals and postcursors, each of which is stacked along their respective incoming directions (Fig. 3C, F). The long-period $S_{diff}$ stacked signals (Fig. 3A) look similar to unstacked waveforms (Fig. S5), while the short-period postcursor stacked signals (Fig. 3D) emerge from what appears to be mainly background noise in the raw data (Fig. S5).

We observe that at long-periods the postcursor arrives at around 35 to 50 s, varying with azimuth (Fig. 3B). Strikingly, at short-periods the postcursors are significantly more delayed, arriving at 50 to 70 s (Fig. 3E). Frequency-dependent travel times with differences on the order of tens of seconds between long and short-period $S_{diff}$ postcursor waves have never been documented before, though weak frequency-dependent dispersion of core-diffracted waves has been suggested based on global statistical analysis of ray parameters[20]. The main $S_{diff}$ phase and the postcursor have different incoming directions due to 3D out-of-path effects (Figs. 2, S7 and S8). While the incoming backazimuth direction of $S_{diff}$ phases is slightly scattered (Fig. 3C, F), they remain reasonably consistent across both long- and short-period measurements. Gradual deviation from 0° to 15° away from the epicenter backazimuth (Fig. 3C, F) implies a strong bending effect that can only be explained by interaction with a structure of strong velocity contrast[7]. Combining observations of postcursor travel-time delays and backazimuth deviations, suggests that while waves at the two periods sample similar geographical regions, the seismic velocities at the different length scales above the core–mantle boundary they are sensitive to, differ significantly.

**Modeling of the ULVZ's internal structure**. Detailed waveform modeling of the Hawaiian ULVZ based on long-period postcursor data has been shown in the previous study[7]. We refine their preferred model of a 20 km tall cylindrical ULVZ of radius 455 km with a shear velocity reduction of 20%, to include details of finer internal structure based on our short-period observations and updated location. Initially we explore the model space of a simplified cylindrical ULVZ using computationally cheap ray-based modeling (Figs. S11–S13), followed by 2.5D modeling[19] down to 3 s ("Methods"). This reveals that the short-period postcursor observations can be explained by an ~2 km thick layer with extreme velocity reduction (40%) at the base of the ULVZ, or by the presence of a less anomalous, but wider spread, basal layer (Fig. S14).

While it is still very challenging to simulate full 3D ULVZ synthetics at the high frequencies we explore here, a recent method development combining wavefield injection with Axi-SEM3D makes full 3D global ULVZ synthetics down to 1 s achievable for the first time[21]. We compute 3D synthetics using a 20 km thick ULVZ model with an extreme 2 km basal layer of 40% shear velocity reduction, based on our 2.5D modeling (Mesh details in Fig. S15). We also compute three additional models for comparison (Fig. 4C): a uniform ULVZ with a shear velocity reduction of 20%, a gradient ULVZ varying from a shear velocity reduction of 10% at the top to 30% at the bottom, and a two-layered ULVZ with an upper layer of 10 km reduced by 10% in velocity and a bottom layer of 10 km reduced by 30% (Fig. 4C). The latter three models have an equivalent velocity reduction when integrated vertically across the ULVZ. Long-period data are unable to discriminate between these four models (Fig. 4A). Short-period data however, show strong differences in the travel times of the modeled $S_{diff}$ postcursors (Fig. 4B), with the uniform ULVZ model showing the shortest delay times and the two-layered model showing the longest delay times. Both the gradient ULVZ and the ULVZ with a 2 km extreme basal layer show good fits to the observed dispersion across both frequency bands. While not presenting a unique solution, these 3D high-frequency synthetics demonstrate that a strong vertical variation and extreme velocities at the base above the core–mantle boundary are required to explain the observed waveform dispersion.

## Discussion

This unprecedented record of extreme seismic velocity reduction in the basal layer of the Hawaiian mega-ULVZ (Fig. 5) sheds new light on the nature of these features and the complex processes

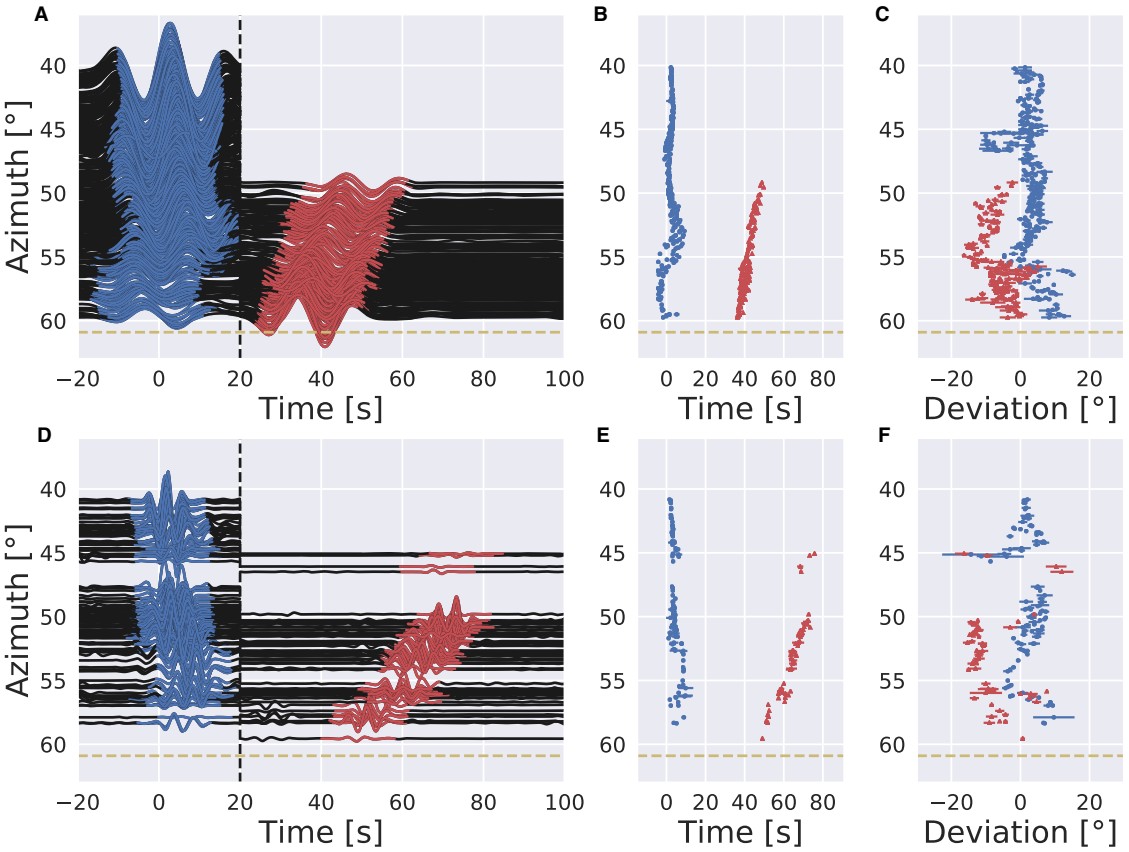

**Fig. 3 Stacked data and beamforming results for the main Sdiff phases and postcursors arrivals of the 20100320 event.** The time axis is aligned by the $S_{diff}$ travel time predicted by PREM[49]. The beamforming is performed by the predicted $S_{diff}$ slowness at 8.32 s/°. **A** Phase-weighted stacks for 10–20 s filtered data. Data stacks before 20 s (black dashed line) use the incoming direction of the main arrival (blue), post 20 s stacks use the incoming direction of postcursor energy (red). **B** Travel time measurements from the maxima of stacked waveform envelopes in (**A**). **C** Backazimuth deviations, i.e. the incoming directions used in the stacks. Error bars represent the max variance of the measurement when the value of $S_{diff}$ slowness is varied by ±5%. **D–F** as for (**A–C**) but for 3–6 s filtered data. Yellow dashed lines show the azimuth through the center of the proposed ultra-low velocity zone.

happening at the core–mantle boundary. The steep thermal boundary just above the core is likely to explain some of the change in velocity with depth, but thermal effects can, at most, only explain several percent velocity reduction. The approximately axisymmetric shape of the ULVZ suggests a natural dynamical link between a cylindrical plume upwelling and the partial melting that might occur at its base[9]. However, the extreme shear velocity reductions of up to 40% that we observe would require a melt fraction likely to be above the percolation threshold[22]. Additionally, these melts would likely be iron-rich and denser than the solid[23], causing them to drain out[22,24].

Our velocity constraints instead suggest a compositionally distinct mega-ULVZ containing increasing iron content with depth. Iron-rich post-perovskite[25] or iron-rich magnesiowüstite[14] have been proposed as candidate minerals that could form significant proportions of solid-state ULVZs. Mixing iron-rich magnesiowüstite with bridgmanite would suggest a model with ~20% magnesiowüstite at the top to ~70% magnesiowüstite in the extreme basal layer[14].

ULVZs have been hypothesized to represent remnants of an ancient global basal magma ocean[26]. If this is the case, the vertical varying ULVZ structure observed holds clues to changing levels of iron fractionation at different stages of magma ocean crystallization. Alternatively, strongly iron-enriched compositions in the lowermost mantle may trace Earth's early impact history, when iron-rich impactor cores mixed with the silicate mantle and accumulated at the core–mantle boundary[27]. An alternative source of iron enrichment is through interactions with the Earth's

core. The extreme anomalous composition on the CMB might allow compositional mass transfer from the core[28]. It is possible that the extreme basal layer in our model represents the first observation of crystallization products formed by core exsolution —a process that is suggested to have driven the early dynamo[29–31]. Core products infiltrating the mantle is supported by observations of anomalous, potentially core-related, isotopic signals in hotspot lavas[15,16,32], which imply the Hawaiian ULVZ is not a closed reservoir, but is likely to be entrained in small amounts into the plume[33].

The observation of this extreme anomaly in the deepest mantle, not only pushes the boundary on the degree of chemical heterogeneity present, but also implies a strong variation in heat flux across the CMB, suggesting greater potential for mechanical and electromagnetic coupling[28]. These are important boundary conditions needed to understand the convective flows generating the geodynamo and mantle plumes[34–37].

## Methods

**Data processing**. All the data for this study are obtained from the IRIS Data Management Center. We select seven events with depths from 12 to 413 km and seismic moment magnitude from 5.9 to 7.8 that sample the Hawaiian ULVZ. The detailed source information is shown in the Table 1. We remove the instrument response by convolving the original data with the inverted instrument spectrum using ObsPy[38]. Horizontal components are rotated to the radial and tangential orientations. In this study we analyze the tangential component of the seismogram, because the SH component of the $S_{diff}$ wave propagates further and attenuates less than the SV component and thus has stronger energy. Low-quality noisy traces are manually identified and removed. We use a zero-phase fourth-order Butterworth

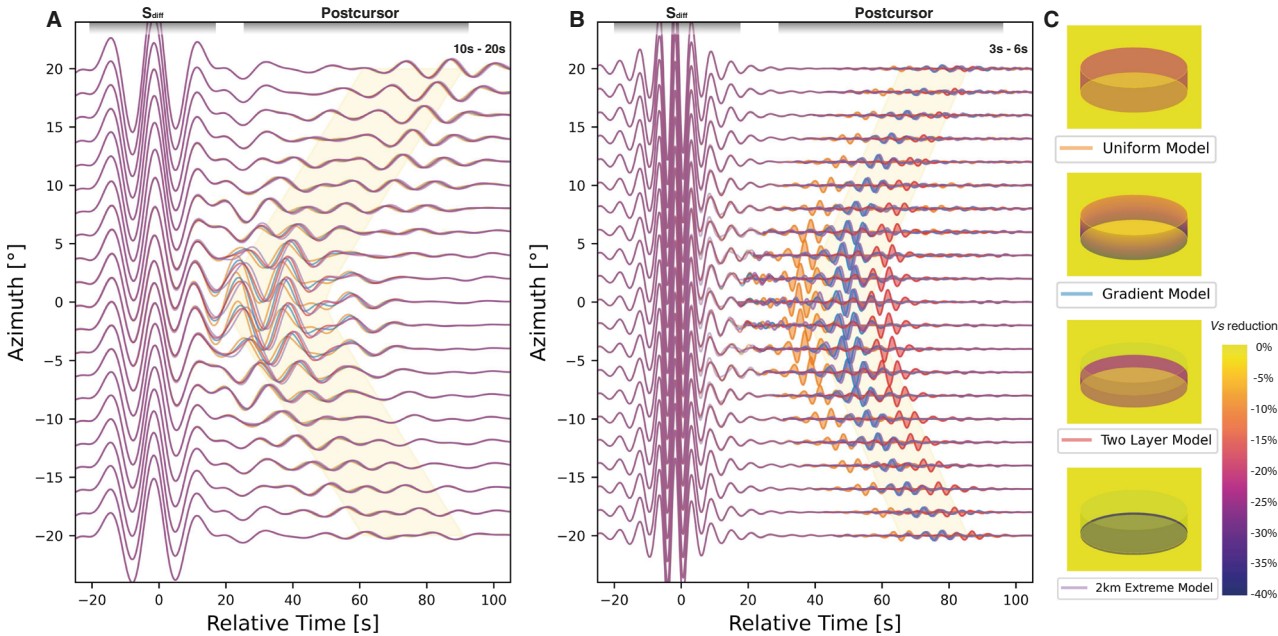

**Fig. 4 3D Synthetic waveforms of four different ultra-low velocity zone (ULVZ) models computed down to period of 3 s.** Synthetics are made for a uniform ULVZ model of 20 km with −20% $dlnV_S$ (orange), a two-layer ULVZ model of 10 km −10% $dlnV_S$ layer on top plus a 10 km −30% $dlnV_S$ layer at bottom (red), a 20 km gradient model changing gradually with ULVZ height from −10% to −30% $dlnV_S$ (blue), and an extreme 2 km −40% $dlnV_S$ basal layer (purple) within a ULVZ of 20 km thickness with −20% $dlnV_S$. Synthetic waveforms filtered into the (**A**) long-period range (10–20 s) and (**B**) short-period range (3–6 s). The source depth and the distance to the ULVZ are the same as for Event 20100320. The ULVZ model is located at 0° azimuth and 39.7° distance from the source. The synthetics are shown for epicentral distances of 105° as a function of azimuth. The time axis is aligned by the $S_{diff}$ travel time predicted by PREM[49]. The waveforms of the short-period postcursor are filled to emphasize them. Note that the gradient model (blue) and 2 km extreme basal model (purple) nearly overlap. The faded yellow shading indicates the time delay ranges for the long- and short-period postcursors as observed in the real data. (**C**) Cartoons of four different ULVZ models tested in the synthetics.

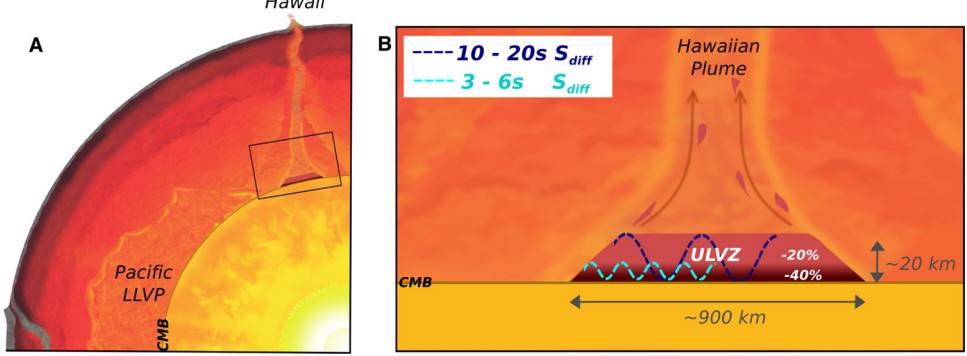

**Fig. 5 Conceptual cartoons of the Hawaiian ultra-low velocity zone (ULVZ) structure. A** ULVZ on the core–mantle boundary at the base of the Hawaiian plume (height is not to scale) (**B**) a zoom in of the modeled ULVZ structure, showing interpreted trapped postcursor waves (note that the waves analyzed have horizontal displacement).

**Table 1 Events used in this study. Source information is obtained from the global CMT catalog.**

| Event | Region | Date and time (UTC) | Latitude (°) | Longitude (°) | Depth (km) | Magnitude ($M_W$) |
|---|---|---|---|---|---|---|
| 20100320 | New Ireland Region, P.N.G. | 2010-03-20 14:00:50 | −3.32 | 152.33 | 413 | 6.6 |
| 20111214 | Eastern New Guinea Reg., P.N.G. | 2011-12-14 05:04:59 | −7.49 | 146.83 | 133 | 7.1 |
| 20120417 | Eastern Neaw Guinea Reg., P.N.G. | 2012-04-17 07:13:49 | −5.66 | 147.16 | 209 | 6.8 |
| 20180518 | South Of Kermadec Islands | 2018-05-18 01:45:31 | −34.67 | −178.21 | 14.3 | 6.1 |
| 20181030 | North Island, New Zealand | 2018-10-30 02:13:39 | −39.07 | 174.94 | 226 | 6.1 |
| 20161122 | North Island, New Zealand | 2016-11-22 00:19:43 | −40.79 | 177.58 | 12.0 | 5.9 |
| 20180910 | Kermadec Islands Region | 2018-09-10 04:19:02 | −31.91 | −179.13 | 119 | 6.9 |

(http://www.globalcmt.org/). Event 20100320 shows the strong dispersion in the postcursor presented in the main paper.

band-pass filter for filtering. Examples of the tangential component S$_{diff}$ data for the 2010 event filtered in 10–20 s and 3–6 s period bands are presented as a function of azimuth are shown in Fig. S5.

**Update on location of Hawaiian ULVZ.** The previous modeled location of the Hawaiian ULVZ sits just southwest of the surface hotspot, centered roughly between 172.5 W and 162.5W[7]. With the diffracted data from events in Papua New Guinea recorded at the transportable array in the central USA, which propagate mainly from west to east, the ULVZ location was well-constrained latitudinally (Fig. S2), but a degree of uncertainty remained as to the exact longitudinal location due to lack of crossing data at the time of publication. The recent deployment of the Alaska transportable array provides a new direction to illuminate the Hawaiian ULVZ using diffracted phases propagating from south to north. We identify four new events from the Kermadec Islands that are recorded in Alaska which show similar hyperbolic S$_{diff}$ postcursors to the previous events recorded in the central USA (Fig. S2). The symmetry of the postcursors suggests the Hawaiian ULVZ has an axisymmetric structure likely quasi-cylindrical in shape. In this case, the least delayed postcursors represent waves that have propagated through the center of the cylinder have not been refracted out of plane, and thus show no additional travel-time delay due to extra path length. Based on the intersection of the minimally-delayed S$_{diff}$ paths in two almost orthogonal directions (using 20100320 and 20180910 events), the ULVZ is located further to the southwest than previously thought, centered around 172.3°W and 15.4°N (Fig. S3). Long-period synthetics for this new location are computed using the coupled spectral-element method CSEM[39] and compared to data from all events, as shown in Figs. S2 and S3. Results show a good fit to the delay time and move-out of postcursors, although there are some variations between the observed and modeled amplitudes. The discrepancies between main phases and depth phases are likely due to inaccuracies in the CMT sources[40] and the S$_{diff}$ radiation patterns. The reduction of the main phase energy is largely underestimated in the synthetic data.

While some of these events have a hint of a high-frequency further delayed postcursors, none were of convincing quality to allow interpretation. However, the overall noisy data and presence of depth phases, do not allow us to definitively confirm the absence of high-frequency postcursors from this dataset.

**Sensitivity kernel for S$_{diff}$ wave at CMB.** The sensitivity kernel of a wave illustrates which part of the Earth affects the observed waveform. Some numerical software packages (e.g. SPECFEM3D_GLOBE[41,42]) can now calculate the finite-frequency sensitivity kernels for specific phases. However, calculations for kernels at higher frequencies (i.e. up to 0.33 Hz) are still very challenging given current computational resources. Here we approach the high-frequency sensitivity kernels with a more heuristic analytical method. We note that the tangential component of the guided postcursor shear waves we observe near the CMB are analogous to surface Love waves, as they both have free stress boundaries in the SH system. We apply the theory previously developed for Love waves in a vertically heterogeneous medium[43] to velocity profiles at the CMB in order to provide an estimate of the S$_{diff}$ sensitivity kernel at different frequencies. We assume the wave coming from the mantle side is a quasi-plane wave of a specific frequency and wavenumber. We then extend the wave propagation from the lower 200 km to the CMB using the propagator matrix method. Making use of the boundary condition at the CMB, we obtain the eigen wavenumber for each specific frequency and then transform them into a sensitivity kernel. Fig. S4 shows the eigensolutions of displacement and traction for 3-s, 10-s, and 20s-period waves. From the plot, we see that shorter periods have sensitivity closer to the CMB, with the 3s-period showing significant sensitivity to the lowermost 5 km. These kernels have guided our proposed velocity models used to fit our multi-frequency observations.

We also estimate the first Fresnel zone widths by computing the width at the core–mantle boundary for which arrivals at the stations will constructively interfere (i.e. arrive within wave period/4). For a 10–20 s period we estimate the half width to be 200–300 km, and for a 3–6 s wave 100–200 km. Note that these widths are less than the radius of the ULVZ.

**Beamforming analysis.** Beamforming is an array method used to measure the incoming direction and slowness of a signal as it passes through an array[44]. We use beamforming not only to determine the direction of the incident wave, but also to enhance the energy of the original signal by stacking data using the measured incident direction to align specific phases. The beamforming stack (B) is given by:

$$B(t, \theta) = \text{abs}[H(\sum_{i=1}^{N} s_i(t - u(\theta) \cdot x_i))] \qquad (1)$$

where $H$ represents the Hilbert transform on the stacked original data series $s_i(t)$, $u$ is the slowness vector as a function of the incoming angle $\theta$, and $x_i$ is the distance vector to the reference station.

First, we apply this beamforming stacking on the original array data in order to determine the most likely incoming angle. Unlike other body waves, the S$_{diff}$ phase has a fairly constant slowness value for the grazing distances as long as the velocity variations are negligible at the diffraction exit points at the core–mantle boundary. We fix the absolute value of S$_{diff}$ slowness $u$ at 8.323 s/deg, calculated in the IASP91

model[45], and search the incident direction in a range of −50 to 50 degrees with respect to the incoming angle of the great circle path from the event epicenter provided by the TauP software[46]. We apply this procedure for each station by forming a subarray consisting of the nearest 20 stations, or all stations within 4 degrees epicentral distance.

Figure S5 shows examples of the resulting beamforming phase stacks. We apply an objective automatic picking procedure, which finds the coherent energy peak in the time range for main arrival and postcursor respectively. The pick is kept only if we observe one and only one dominant local maximum in the estimated arrival time window. The backazimuth directions and travel times for the maxima in our beamforming stacks are shown in Fig. 3BC and EF in the main text.

We note that 3D out-of-path effects and velocity variations at the turning point of the seismic ray have the potential to cause slowness variations leading to significant mislocation in plotted slowness vectors. To test whether this has an impact on our beamforming results we perform a sliding-window F-K analysis of the S$_{diff}$ main arrival and postcursor allowing slowness to vary from predicted values. An example of this test using a subarray of multiple stations surrounding seismic station TA.T33A is shown in Fig. S8. We find that although the absolute slowness value of phases is slightly off the predicted 8.3 s/deg, almost all the values lie within a range of ±5% (±0.4 s/deg). Thus, we repeat the procedure with slowness values varied by ±5% to estimate the uncertainty of our fixed slowness. The differences in measured backazimuth and travel time are shown as error bars on our measurements in Fig. 3BC and EF.

We also retested our beamforming result allowing for greater variation in slowness of ±10% (slowness differences 0.8 s/deg) and compared this with our initial ±5% slowness variance calculations, as shown in Fig. S9. Although we miss some data points when using a ± 10% variance (this might be due to the multiple arrivals or failure to locate the peak in automatic scripts), we see the backazimuth beamforming of the S$_{diff}$ signal returns highly similar results, compared to our original ±5% variance analysis. We find that while there is some small tradeoff between the backazimuth, arrival time, and the absolute slowness value, the influence of absolute slowness has a minimal influence on results.

**Waveform stacking.** We produce final waveform stacks by multiplying the linear stack with a phase-weighted stack using the observed incidence angle measurements of the main phase and postcursor, such that

$$g(t) = \frac{1}{N} \sum_{i=1}^{N} s_i(t - u \cdot x_i) \| \frac{1}{N} \sum_{k=1}^{N} \exp[i\Phi_k(t - u \cdot x_i)] \|^{\nu} \qquad (2)$$

where $s_i$ represents the time series of $i$th station of an array, $x_i$ is the distance vector for $i$th station with respect to the reference station, $u$ is the measured directional slowness vector for the reference station obtained from beamforming, and $\Phi_k$ denotes the instantaneous phase obtained from Hilbert transform of the original data series $s_i(t)$. Weighting factor $\nu$ governs the sharpness of the noise reduction. To balance data distortion and signal coherency, we use $\nu = 2$ in our processing, as recommended in ref. [47]. We again stack over sub-arrays where $N = 20$ nearest stations, or all of the stations within four degrees epicentral distance.

Because of the differences between our main arrival and postcursor phases, we have to split our stacked seismogram into two parts: the first window uses the measured main arrival incoming direction and the second window uses the measured postcursor incoming direction. The final stacked waveforms for all stations are shown in Fig. 3A, C in the main text, with the two parts split by a dashed line. This procedure is applied for the two different frequency ranges explored, and significantly aids in bringing out a coherent postcursor signal in the 3–6 s period range. For comparison to the stacked waveforms shown in Fig. 3, the raw data of event 20100320 are shown in Fig. S5.

**Wavelet spectrum of the stacked seismograms.** The dispersive nature of the observed postcursors is difficult to observe in seismograms in the time domain. Here we use the wavelet spectrum, which is well-suited for non-stationary seismic data, to show the energy distribution of signals in both the frequency domain and the time domain[48]. These spectra illustrate the stronger phase dispersion in the postcursor compared to the main S$_{diff}$ arrival. We perform a continuous wavelet transform using the complex Morlet wavelet on the stacked time series from 20 s before to 120 s after the predicted S$_{diff}$ arrival time. Spectra of linear stacks and phase-weighted stacks based on main S$_{diff}$ and postcusor incoming angles are compared in Fig. S10. The phase-weighted stacks have a better signal-to-noise ratio. The spectra show that most of the signal energy is distributed above 3 s. We also observe a gap in energy near the period of 6 s in the data from event 20100320. It is based on these observations, that we define the two frequency ranges used in this study to demonstrate the dispersive nature of these phases: short-period at 3–6 s, and long-period at 10–20 s.

**First estimate of basal ULVZ properties.** Although calculating wave propagation through a 3D ULVZ model is a complicated process, we first estimate the basic properties of the base of the ULVZ using simple calculations of postcursor time delay. Along profile that samples the ULVZ center, multi-pathing effects are negligible. This provides us with a simple geometrical relationship that links the properties of the ULVZ with the arrival time delays:

$$\delta t \sim \frac{2r}{v} \frac{dv}{1 - dv} \qquad (3)$$

where $r$ is the radius of the structure, $dv$ is the fractional velocity reduction, and $v$ is the background velocity at the CMB. This relationship is plotted in Fig. S9 for the least delayed long-period and short-period postcursors. Results reveal that either extremely slow material at the base of the ULVZ or a ULVZ that is much wider at its base could explain the arrival times of delayed postcursors. If the bottom part of the ULVZ remains the same radius at 455 km, the velocity reduction would be up to 30% (Fig. S11). If we otherwise assume the velocity reduction to remain constant at 20%, then the material should be spread much more widely with a radius of 700 km.

**Ray-based modeling of the ULVZ**. Next, we use an approximate ray-based method and the travel-time and backazimuth constraints from the refracted short-period postcursors to further estimate the properties of the lowermost part of the ULVZ structure. We only predict the horizontal wave propagation using the horizontal slowness and assume a full decoupling of the vertical propagation. The transmission of the wave is calculated using Snell's law only where the ray enters and exits the ULVZ boundary. Figure S12 shows the refraction pattern caused by the cylindrical ULVZ and the predicted travel times and backazimuths as a function of azimuth from the event. Note that the real wavefield is more complicated than this simplified modeling method predicts, since it assumes infinite frequency rays and there is no height specified for the ULVZ. In the full waveform wavefield, some energy, particularly at longer periods, will propagate over the ULVZ. Thus, we observe some main $S_{diff}$ energy in the real data from 50° to 65° azimuth which is absent in this simplified simulation that only provides a first order estimate of $S_{diff}$ postcursor behavior.

With this computationally efficient tool, we can explore a large parameter space of radius and velocity reductions for the ULVZ, before computing more expensive full waveform synthetics. We implement a grid search varying the ULVZ velocity reduction from 15 to 50% in steps of 5%, and its radius from 355 km to 855 km in 50 km steps. Figure S11 shows how well different combinations of ULVZ properties fit the short-period postcursor observations. Misfits are calculated based on the arrival times, the backazimuths, and a combination of the two. We choose to use the L1 norm in the misfit calculation, as it is more robust and resistant to outliers for a small number of samples.

In the travel-time misfit plot (Fig. S13A), we see the expected tradeoff between the velocity reduction and the ULVZ radius. The travel-time misfit suggests the cylindrical ULVZ to be either small, with an extreme velocity reduction (radius 405 km, $-50\%$ $dlnV_S$), or to be larger, with a more modest velocity reduction (radius 855 km, $-20\%$ $dlnV_S$). The backazimuth misfit also shows a tradeoff, but with the opposite orientation (Fig. S13B): requiring the ULVZ to be either small with slight velocity reduction (radius 455 km, $-15\%$ $dlnV_S$) or to be widespread and extremely reduced (radius 855 km, $-45\%$ $dlnV_S$). The combination of these two misfits normalized by each minimum value, creates a joint misfit which employs the two opposing trade-offs. This misfit identifies the best-fitting intermediate model with a radius between 605 km and 755 km and a velocity reduction between 25 and 30% (Fig. S13C). The best-fit result peaks at radius 655 km and velocity reduction at 30%. Misfits for a ULVZ of a constant radius of 455 km are less good, but would predict a velocity reduction of 40–45%.

**2.5D synthetic exploration of basal layer thickness**. We build on initial ray-tracing based results to further explore the causative effect of the ULVZ lateral structure using AxiSEM 2.5D synthetics down to 3s[18]. These synthetics allow us to capture the full finite-frequency sensitivity of the diffracted phases but are more computationally feasible in terms of exploring the model space than full 3D simulations. 2.5D models assume azimuthal symmetry and are only accurate in the event-station plane, such that out-of-plane effects cannot be captured. Thus, we compare our synthetic results to the waves that have propagated through the center of the ULVZ without refracting, i.e. the postcursors waves with minimal delay times. These waves show a travel-time deviation of 12 s between the long- and short-period postcursors. We construct two end-member models for the mega-ULVZ structure: (1) a 455 km radius mega-ULVZ model with an extremely reduced basal-layer of $-40\%$ $dlnV_S$, referred to as R455; 2) a similar-sized ULVZ including a 655 km wide and $-30\%$ $dlnV_S$ basal-layer reflecting the lateral spreading of the mega-ULVZ material, referred to as R655. We analyze how the height of the basal layer in both models interacts with $S_{diff}$ waves by varying its thickness from 0 to 5 km (Fig. S14). We find that for model R455 the best fitting solution suggests a 2 km thick extreme basal-layer, while for model R655 a basal-layer thickness slightly above 2 km is suggested.

**Full 3D $S_{diff}$ synthetics**. Using our estimates of ULVZ properties from the above analyses, we create models for full 3D waveform modeling. A new hybrid method has recently been developed using a combination of wavefield injection and Axi-SEM3D, which allows computation of the global wavefield with small-scale 3D internal heterogeneity down to periods of 1s[21]. This method stores the wavefield information at the injection boundary and then uses AxiSEM3D to compute the wavefield inside and outside the boundary. As demonstrated in the 2.5D synthetics, our $S_{diff}$ observations are more sensitive to vertical rather than lateral structure. Thus, we explore several variations of the R455 model with internal vertical structure. With a dense vertical mesh to capture our input model on the kilometer

scale, each of such synthetics takes 128 nodes, with 64 cores per node, 96 h to finish on the Cambridge HPC cluster. Testing the wider R655 ULVZ scenario would require a larger injection boundary, and thus even more computing resources.

In Fig. S15, we show a detailed AxiSEM3D mesh implemented in the 3D waveform modeling. The mesh is generated for PREM model at a 3 s period. The mesh consists of 731,418 spectral elements, with each formed by 25 GLL points at a polynomial order of 4. At the bottom 20 km of the mantle, the mesh density is doubled to accommodate any shear velocity reduction up to 50%, and a vertical discontinuity at 2 km, 6 km, 10 km or 15 km above CMB can be accurately honored (because the 20 km height contains five element layers).

We construct four different ULVZ models for comparison: a uniform ULVZ of $-20\%$ $dlnV_S$, a gradient ULVZ varying from $-10\%$ $dlnV_S$ at the top to $-30\%$ $dlnV_S$ at the bottom, and a two-layered ULVZ with an upper 10 km at $-10\%$ $dlnV_S$ and a bottom 10 km at $-30\%$ $dlnV_S$, and a 20 km thick ULVZ model with an extreme 2 km basal layer of $-40\%$ $dlnV_S$ based on our 2.5D modeling. Figure 4 in the main text shows the waveform filtered into the two period bands of interest. The 10–20s waveforms (Fig. 4A) are highly similar for all models, demonstrating that waves at these periods are unable to differentiate between models containing different kilometer-scale internal structure. Data filtered for the 3–6s band (Fig. 4B) show strong differences between the modeled waveforms. The uniform model has a postcusor around 40 s after the main $S_{diff}$ phase. The postcusor for the two-layer model arrives much later at roughly 60–70 s. The postcursor of the gradient model and the 2 km extreme model overlap at arrival times around 50–60 s—similar to arrivals seen in our high-frequency observations. All postcursors in these four models show a dispersive nature, to varying extents, between 3 s and 10 s (Fig. S16). The uniform and two-layer models show the weakest and strongest dispersion respectively, while the gradient and 2 km extreme model show moderate dispersion, comparable to our observations. It should be noted that the main $S_{diff}$ phase, unlike its associated postcursor, is not dispersive.

## Data availability
The raw seismic waveform data are archived and available from the IRIS (Incorporated Research Institutions for Seismology) data center (https://www.iris.edu/). The processed waveform data and 3D waveform synthetics generated in this study have been deposited on https://zenodo.org/record/4911586 under public access.

## Code availability
Code of simulating the 3D synthetics waveform modeling in this study is available at https://github.com/kuangdai/AxiSEM-3D. Computer codes used to produce these figures will be made available upon request to the corresponding author.

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

## Acknowledgements

The authors acknowledge the IRIS (Incorporated Research Institutions for Seismology) data center for providing all the data. This work was performed using resources provided by the Cambridge Service for Data Driven Discovery (CSD3) operated by the University of Cambridge Research Computing Service. We thank D. Al-Attar for input on the kernel calculation, Carl Martin for the wavefront simulations that Fig. 4B are inspired by, and W. Liu for discussions on our interpretation. This work is funded by the European Research Council (ERC) under the European Union's Horizon 2020 research and innovation programme (grant agreement No. 804071-ZoomDeep). K.L. is supported by US-NSF and UK-NERC under joint grant NE/R012199/1 and by US-NSF under grant EAR-1610612.

## Author contributions

Z.L.: Conceptualization, Methodology, Software, Formal Analysis, Writing—Original Draft. K.L.: Methodology, Software. J.J.: Visualization, Writing—Review & Editing, Supervision. S.C.: Conceptualization, Methodology, Writing—Review & Editing, Supervision, Funding acquisition.

## Competing interests

The authors declare no competing interests.
