## [Peer Review File · Nature Communications]

REVIEWER COMMENTS

Reviewer #1 (Remarks to the Author):

This is a very interesting study reporting a unique and intriguing observation of dispersion of Sdiff waves and postcursor signals propagating through the Hawaiian ULVZ. It is to my knowledge the first observation of this kind. The study goes further, since the authors also provide various numerical simulation results for modelling their observation using state-of-the-art methods. It is very demanding to generate high frequency waveforms and this effort also deserves to be acknowledged. I found quite clear why "they advance our understanding of the core-mantle boundary compared to earlier work". The implications are also well described. To my point of view the authors have taken into account the editor's and reviewers' suggestions and if not they justify their choice. I am fine with their reply to the reviewer. So for me this paper fits for publication in Nature Communication. I have only one major comment that needs to be addressed, otherwise it is just a matter of some additional clarifications needed in the text or minor changes in the figures.

Major comment:

Figure S10 is very confusing for me. First the authors should give more details about how this simulation has been performed. Second it does not agree with what is drawn figure 4B. Figure S10, if I well understand it, shows that from 45° to 65° azimuth there are only postcursors, the main Sdiff is absent. It is also what is shown on the map, stations behind the ULVZ will only receive rays from within the ULVZ and no main Sdiff. So it seems that there is a problem with the simulation, and that some part of the wavefield is not properly modelled ? This figure is even more confusing since it produces results that do not agree with the observations (figure 2 the main Sdiff is always observed, it does not disappear from 45° to 65° azimuth ?). So more details are needed to better understand this figure S10 and the difference with previous ones.

Minor comment:

Figure 4B is nice but it shows that the main Sdiff arrival is also delayed by the structure, so it could also be measured and used to constrain the ULVZ velocity ? Related to that, it is not clear what the author mean by "The time axis is aligned by the predicted Sdiff traveltimes" (in Figure captions 2, 3 and S2), predicted in which model ? PREM ? In figure 2B and 2E and 3 we do not really see the effect of the ULVZ on the main Sdiff phase, which should also be delayed like it is shown on figure 4B ? Can the author clarified that point.

Lines 62-65 : "Higher frequency Sdiff energy gets trapped in thin mega-ULVZs, becoming delayed and refracted. On a seismogram these guided waves appear tens of seconds after the main Sdiff phase, and for a cylindrical ULVZ show a hyperbolic delayed move-out as a function of azimuth. We refer to these as Sdiff postcursors.". The problem is that "higher frequency" is not very specific and what is defined as "higher frequency" is a matter of Sdiff wavelength compare to the ULVZ lengthscale. It would thus be clearer to describe the phenomena in terms of wavelength. Something like "Sdiff with wavelengths of the order of the ULVZ size or features gets trapped in thin mega-ULVZs, becoming delayed and refracted." So depending on the size it will give you the frequency at which you start to see the postcursors and that will be what you refer to as "higher frequency".

A general comment. When one measures traveltimes and wants to deduce the structure from them, there is a strong trade-off between the velocity perturbation and the size of the anomaly. Here this trade-off is reduced using higher frequency data. However, from figure 3 it is shown and written lines 158-159 that the data cannot distinguish between a gradient model or a model with an extreme velocity reduction at the base of the ULVZ. However, the author kind of focus on the second option only (line 26, 171) which I feel is a bit abusive. So I would recommend to state it the other way around. I mean instead of saying that thanks to the high frequency data they can image an up to -40% velocity reduction at the base of the ULVZ, I would say the data exclude a uniform model or a two layered model with a 10km thick with a 10% velocity reduction on top of a 10km thick 30% velocity reduction. Actually what would be interesting is to test different thicknesses of the bottom layer and see when it agrees with the data. This way a constrain on the maximum size of the extreme bottom layer thickness could be put.

While figures S2 and S3 show postcursors on the data from all the events plotted in Figure 2 only one (20100320) is used in the paper (stated lines 105-106). Why is that ? Postcursors at high frequency for the others events are not much delayed ? or are absent ? Can the authors comment on that and better justify why they only focus on that one event. If postcursors at higher frequency are also delayed for the other events then they should be used in the study. Otherwise their absence should also be understood with the modelling.

Line 401-406: Could the paths with the least delayed postcursors be plotted on the

map for each event so that we can see how they intercept ?

Figure 1: figure 1A and 1B should be exchanged since in the text the authors first refer to 1B. Then it is very hard to see the yellow paths in figure 1B. Either the figure should be enlarged or the yellow color should be avoided ?

Figure 2: I know that this figure is already busy but I would put figure 4B in here because it helps to better understand what is the origin of the postcursors. Otherwise the notion of postcursors is only explain in the text but in a quite vague way while figure 4B is very clear and it precisely illustrates the phenomena. Either it should be a figure by itself or included in figure 2.

Figure 3: 3C is quite small and hard to read. I would put a box around each model case with the color of the waveform that is used in 3A and 3B. And may be it should appear below the waveform in a row instead of a column so that 3C could be enlarged.

Figure 4: it is a very nice one however 4A is quite large compare to 4B and 4C which are really related to the results of this study. So as I said before I would move 4B into figure 2 and actually enlarge 4C and reduce 4A. For instance 4A could focus on the Hawaian plume , the rest of the picture is not needed, even not discussed in the text. What is the gray line that appears in the top left corner of 4A for ?

Figure S1: it is strange because on the raw data we clearly see the disappearance of the main Sdiff between 55-65° at long period and the inverse for the postcursors. And this is no longer true after the stacking (Figure 2). Is that a problem with the stacking ? Figure S1 should be after Figures S2 and S3 since they appear first in the text.

Figure S3B: the data are very noisy and honestly I would not be able to pick any Sdiff or postcursors. In general from figures S2 and S3 I am not sure I understand on what arguments is based the drawing of the yellow bands. Can the author give more details in the caption on how they obtain the yellow bands.

Figure S4: In the Vs kernel panel, the x-axis should go from 0 to 1.

Typos:

line 179: bridgmenite – > Bridgmanite

line 391-392: "The previous modeled location of show Hawaiian ULVZ sits just southwest of the surface hotspot, centered roughly between 172.5W and 162.5W" – > something is wrong in this sentence.

Equation after line 524 is not well displayed

Reviewer #2 (Remarks to the Author):

Li et al. present a study of the mega-ULVZ beneath Hawaii using Sdiff waveforms and perform synthetic modeling to refine the previously-published model of the structure (Cottaar and Romanowicz, 2012). The authors update the location of the ULVZ by utilizing data acquired by the transportable array in Alaska which allowed them to expand their analysis with improved data coverage. The authors also identify frequency-dependent travel times of the Sdiff postcursors and discuss possible small-scale structures associated with the Hawaiian mega-ULVZ.

I find this manuscript interesting and a potentially important contribution to our understanding of the mega-ULVZ beneath Hawaii. The novelty of the work is (1) identification of the high frequency Sdiff postcursors, (2) an update on the location and (3) physical properties of the previous model. I also find that the authors have fully addressed the comments and concerns raised in a previous round by the editor and two reviewers. In my opinion, the manuscript is well-written and all of the figures are publication-ready. I have a few minor points worth further clarification before the manuscript is suitable for publication.

<Introduction>

The authors define "Sdiff postcursors" as seismic waves that arrive after the main Sdiff phase which show a hyperbolic delayed moveout as a function of azimuth, as expected in the case of a cylindrical ULVZ model (e.g., L63-65). Then the authors state that observations of Sdiff postcursor are quite rare (e.g., L69-70). I think that the definition of Sdiff postcursors and terminology used in the manuscript is misleading because it requires a specific model / interpretation in mind; delayed signals that arrive after the main Sdiff phase, Sdiff postcursors, need not follow a hyperbolic moveout as a function of azimuth, because they can be generated by ULVZs with different morphologies and by other lower mantle structures. In such cases, the "Sdiff postcursors" would be left out of the analysis in this manuscript because they would not necessarily show a hyperbolic moveout. Sdiff postcursors are in fact pervasive and observed on many paths across the Pacific basin (e.g., Kim et al., 2020). That structures capable of producing Sdiff postcursors are widespread is also found using independent datasets (e.g. Thorne et al., 2021). Only in areas of excellent data coverage, and when the ULVZs are unusually large, such as near Hawaii (Cottaar and Romanowicz, 2012) and Iceland (Yuan and Romanowicz, 2017), is hyperbolic delayed moveout as a function of azimuth unequivocally observed. Therefore, I suggest rephrasing the aforementioned text for clarity.

<High-frequency Sdiff postcursors>

In my opinion, one of the key observations presented in this manuscript is the identification of the high-frequency Sdiff postcursors. I agree that these phases can provide strong constraints for determining the fine structure of the mega-ULVZ as discussed in the main text. Therefore, the authors should show more examples of these new observations and document their robustness in the manuscript. Currently the authors only show such high-frequency data from a single earthquake (e.g, L105-106; event 20100320) but also acknowledge that high-frequency postcursors are difficult to observe (e.g., L108-109). Can they include some more data examples that display high-frequency postcursors from other events (similar to Fig. S1)? Maybe using the single event was intentional, and postcursors in other event data are contaminated by other arrivals, e.g., depth phases, but I cannot find any further explanation about the choice of event in high-frequency waveform analysis. I think this verification step is crucial since this is one of the most distinctive elements of the manuscript and an independent new constraint on the detailed structure within the Hawaiian mega-ULVZ.

I understand that the waveforms shown in Fig. 2D are highly processed in order to bring out coherent signals between 40-80s. How do the waveforms look in Fig. 2D if you were to apply a simple linear stack to Fig. S1B? The highlighted band really helps to focus what the authors want me to see, but without the band I would also argue that there are very weak arrivals between 20-40s similar to the long-period postcursors in Fig. S1. Am I dreaming here?

<Modeling of the ULVZ>

As discussed in the manuscript, the waveform modeling approach has tradeoffs across different waveform parameters. The authors mainly focus on three parameters, location / radius / velocity variation, but is there a reason not to include height in the analysis? As shown and discussed in the supplement (e.g., Fig. S2-3; L408-410), the amplitude of the data vs. synthetic waveforms does not show a good fit. The goodness of the fit in this manuscript seems to focus on the backazimuth and delay time of the Sdiff postcursors but the synthetic waveforms generally underpredict the anomaly. Do you think this can be due to a sub-optimal height of your anomaly in the modeling? I doubt that all of the observed amplitude discrepancies can be explained by inaccuracies in the CMT solutions and the radiation pattern (e.g., L409-410). Can this be partially caused by physical properties of the ULVZ that are being underpredicted by your proposed model?

Despite these comments/questions and reservations, I am excited that the authors are updating their previous models with more data in different frequencies. I hope the authors find my comments constructive and helpful.

Response to Reviewers

Reviewer comments are shown in black, with our responses in red. Where responses have led to changes in the manuscript these are **highlighted in bold red font**, and the section, line or figure number in the updated manuscript is referenced.

Reviewer #1:

This is a very interesting study reporting a unique and intriguing observation of dispersion of Sdiff waves and postcursor signals propagating through the Hawaiian ULVZ. It is to my knowledge the first observation of this kind. The study goes further, since the authors also provide various numerical simulation results for modelling their observation using state-of-the-art methods. It is very demanding to generate high frequency wave-forms and this effort also deserves to be acknowledged. I found quite clear why "they advance our understanding of the core-mantle boundary compared to earlier work". The implications are also well described. To my point of view the authors have taken into account the editor's and reviewers' suggestions and if not they justify their choice. I am fine with their reply to the reviewer. So for me this paper fits for publication in Nature Communication. I have only one major comment that needs to be addressed, otherwise it is just a matter of some additional clarifications needed in the text or minor changes in the figures.

We appreciate the first reviewer's kind and encouraging observations. We respond to each of their comments below.

Major comment:

Figure S10 is very confusing for me. First the authors should give more details about how this simulation has been performed. Second it does not agree with what is drawn figure 4B. Figure S10, if I well understand it, shows that from 45° to 65° azimuth there are only postcursors, the main Sdiff is absent. It is also what is shown on the map, stations behind the ULVZ will only receive rays from within the ULVZ and no main Sdiff. So it seems that there is a problem with the simulation, and that some part of the wavefield is not properly modelled? This figure is even more confusing since it produces results that do not agree with the observations (figure 2 the main Sdiff is always observed, it does not disappear from 45° to 65° azimuth?). So more details are needed to better understand this figure S10 and the difference with previous ones.

The reviewer is correct that figure S10 shows slightly different predictions than figure 4B – this is because the two figures are generated using different forward modelling methods.

The simulation in Figure S10 (now Figure S12) is performed using a ray-based method that employs Snell's law at the ULVZ boundary to allow fast computation estimates of the expected delay of Sdiff postcursors only – this method does not include the response of the main Sdiff phase. This is because the ray-based method assumes infinite frequency and an infinite height to the ULVZ, thus energy of all frequencies gets affected. In comparison, in the true full waveform wavefield, some energy, particularly the longer period energy, will propagate over the ULVZ, which causes the general arrival of a main phase and obscures energy that diffracted around the anomaly. The lack of such finite frequency effects is typical in ray-based methods. Since we are applying this simulation only to provide a first order estimation the postcursor raypaths and traveltimes, such simulation can still be useful.

In comparison Figure 4B (now Figure 2) while only a cartoon, is inspired by wavefront simulations using an adapted version of the wavefront tracker used for multi-pathing by Hauser et al. G3, 2008. This also allows for diffracted energy around the anomaly, which is

what is also shown in the full waveform simulations (now Figure 4) and reflects what we see in our data observations (now Figure 4) and reflects what we see in our data observations.

Initially the chosen model in Figure S10 was given as a 'random example', but to avoid confusion, we have remade figure S10 (now Figure S12) using our preferred model at the basal layer which has a 455km radius and 40% vs reduction.

To clarify the limitations of the ray based method, we adapted our text in **line 636-648** to give more details of this method:

“Note that the real wavefield is more complicated than this simplified modeling method predicts, since it assumes infinite frequency rays and there is no height specified for the ULVZ. In the full waveform wavefield, some energy, particularly at longer periods, will propagate over the ULVZ. Thus, we observe some main S_{diff} energy in the real data from 50° to 65° azimuth which is absent in this simplified simulation, which only provides a first order estimate of S_{diff} postcursor behaviour.”

Minor comment:

Figure 4B is nice but it shows that the main S_{diff} arrival is also delayed by the structure, so it could also be measured and used to constrain the ULVZ velocity? Related to that, it is not clear what the author mean by "The time axis is aligned by the predicted S_{diff} traveltimes" (in Figure captions 2, 3 and S2), predicted in which model? PREM? In figure 2B and 2E and 3 we do not really see the effect of the ULVZ on the main S_{diff} phase, which should also be delayed like it is shown on figure 4B? Can the author clarified that point.

The wavefronts in Figure 4B are a cartoon, based on the simulations in the wavefront tracker on a 2D spherical surface [Hauser et al, G3, 2008]. This method predicts energy to diffracted around the anomaly, but does not account for any finite frequency waveform healing effect nor the 3D structure effect of a finite height ULVZ which we see in full-waveform simulations, which are more consistent with observations, as noted by the reviewer for Figures 2B, 2E and 3. These are interesting points raised by the reviewer, but we don't think we can include this level of detail on the cartoon in the paper.

The time axis is aligned by the S_{diff} traveltimes predicted by the PREM model. We have rephrased the caption in Figure 3, 4 and S1 (previous Figure 2, 3 and S2) to make this clearer.

Lines 62-65 : "Higher frequency S_{diff} energy gets trapped in thin mega-ULVZs, becoming delayed and refracted. On a seismogram these guided waves appear tens of seconds after the main S_{diff} phase, and for a cylindrical ULVZ show a hyperbolic delayed move-out as a function of azimuth. We refer to these as S_{diff} postcursors." The problem is that "higher frequency" is not very specific and what is defined as "higher frequency" is a matter of S_{diff} wavelength compare to the ULVZ lengthscale. It would thus be clearer to describe the phenomena in terms of wavelength. Something like " S_{diff} with wavelengths of the order of the ULVZ size or features gets trapped in thin mega- ULVZs, becoming delayed and refracted." So depending on the size it will give you the frequency at which you start to see the postcursors and that will be what you refer to as "higher frequency".

This reviewer's comment very helpful in clarifying the meaning of "higher frequency" we want to address here. We clarified the sentence as follows in **lines 62-64**:

“ S_{diff} energy at higher frequencies and shorter wavelengths (on the order of the ULVZs height) which propagate closer to the core-mantle boundary, can get trapped in thin mega-ULVZs, becoming delayed and refracted.”

A general comment. When one measures traveltimes and wants to deduce the structure from them, there is a strong trade-off between the velocity perturbation and the size of the anomaly. Here this trade-off is reduced using higher frequency data. However, from figure 3 it is shown and written lines 158-159 that the data cannot distinguish between a gradient model or a model with an extreme velocity reduction at the base of the ULVZ. However, the author kind of focus on the second option only (line 26, 171) which I feel is a bit abusive. So I would recommend to state it the other way around. I mean instead of saying that thanks to the high frequency data they can image an up to -40% velocity reduction at the base of the ULVZ, I would say the data exclude a uniform model or a two layered model with a 10km thick with a 10% velocity reduction on top of a 10km thick 30% velocity reduction. Actually what would be interesting is to test different thicknesses of the bottom layer and see when it agrees with the data. This way a constrain on the maximum size of the extreme bottom layer thickness could be put.

We appreciate this comment from the reviewer and agree that there is a strong trade-off between the velocity perturbation and the size of the anomaly. Regarding the bottom layer thickness however, we applied preliminary tests using 2.5D AxiSEM3D synthetics (shown in Figure S14) to investigate possible basal layer thicknesses, as described in the supplementary. Our result shows the thickness of an extreme bottom layer of around 2km fits our data observations best for the two model scenarios tested.

The other constraint on the layer thickness is based on the frequency content. The trapped energy of certain frequency ranges corresponds to certain layer thickness as shown in the kernel (Figure S4). The higher frequency data we observe are sensitive to the basal layer. As the reviewer notes, if we consider a gradient model, the parameter space will grow quickly, making it difficult to explore all potential options, which is not practical given the computational expense of full 3D simulations. Nevertheless, a strong vertical variation and extreme velocities at the base is necessary to explain our data.

We agree that we have been a bit biased towards a layered model in our interpretation. This is mainly because it is easier to explain by compositional layering. We have not found a natural explanation for a gradient of 1% velocity reduction per km (or is the entire ULVZ finely layered).

Nevertheless, we do not feel we overly emphasise the basal layer over the gradient model (which we note still suggests velocity reductions of 30% at the base in the 3D example we test, but could be stronger depending on what point within the ULVZ the gradient begins). In the text and interpretation, we state general but not definitive values, which do not indicate whether extreme slow values need to be concentrated in a basal layer or could be reached through a more gradational change.

e.g. Line 26 **“Results reveal that the lowermost part of the Hawaiian ULVZ is extremely reduced in shear wave velocity, by up to -40%.”**

We have reworded **line 186-189** to better address this point:

“While *not presenting a unique solution*, these 3D high-frequency synthetics demonstrate that *a strong vertical variation and extreme velocities at the base above the core-mantle boundary are required* to explain the observed waveform dispersion.”

While figures S2 and S3 show postcursors on the data from all the events plotted in Figure 2 only one (20100320) is used in the paper (stated lines 105-106). Why is that ? Postcursors at high frequency for the others events are not much delayed ? or are absent ? Can the authors comment on that and better justify why they only focus on that one event. If

postcursors at higher frequency are also delayed for the other events then they should be used in the study. Otherwise their absence should also be understood with the modelling.

We acknowledge that only one event (20100320) is used in this manuscript for the analysis of high-frequency postcursors. At the time of the work, we did look through the events presented in Figures S2 and S3 (now figures S1 and S2), and while we found hints, we did not find convincing postcursors justifying further analysis. However, the level of noise in the higher frequency data (and for some events the presence of depth phases) didn't allow us to definitively confirm the absence of these postcursors either. We have added the following text in the supplementary materials in lines 485-488 noting this:

“While some of these events have a hint of a high frequency further delayed postcursors, none were of convincing quality to allow interpretation. However, the overall noisy data and presence of depth phases, do not allow us to definitively confirm the absence of high frequency postcursors from this data set.”

We do note that since the first submission of the paper (in November 2020), another PhD student in the group, Carl Martin, has much expanded the search and found many new geometries of data with postcursors as well as high-frequency postcursors for the Hawaii (as well as Iceland) ULVZs. These observations will be published as a separate first-author paper by Carl.

Line 401-406: Could the paths with the least delayed postcursors be plotted on the map for each event so that we can see how they intercept ?

This comment from the reviewer is very helpful. It is something we had done, but not shown in the previous manuscript. Based on it we have added an extra figure S3 in the supplementary text showing the paths of least delayed postcursors, which is referenced from the main text in line 97. Our main constraint of the ULVZ location is determined from the paths of Event 20100320 (East-West direction) and Event 20180910 (North-South direction). The events at Papua New Guinea and at Kermadec Islands are close to each other and provide similar constraint from two directions. Although the paths do not intercept perfectly at our pinpointed ULVZ location, from the map we see the match still shows a good agreement at our preferred location (172.3°W, 15.4°N).

Figure S3. Paths with the least delayed postcursors for each event intercept at updated Hawaiian ULVZ location. Azimuth with the least delayed postcursors (shown in black dot dashed lines, 61° azimuth for event 20100320, 61° azimuth for event 20111214, 61° azimuth for event 20120417, 9° azimuth for event 20180910, 11° azimuth for event 20180518, 15° azimuth for event 20181030, 14° azimuth for event 20161122) on the background tomography model SEMUCB_WM1 at 2800 km depth³⁶. Beachballs, S_{diff} ray paths (within 1° to the least delayed azimuth), and station notation same as in Figure 1 (including events 20100320 (yellow), 20111214 (green), 20120417 (red), 20180910 (purple), 20180518 (brown), 20181030 (pink), 20161122 (grey), stations (triangles)). Proposed ULVZ location shown with black circle and centered at the red cross (172.3°W , 15.4°N).

Figure 1: figure 1A and 1B should be exchanged since in the text the authors first refer to 1B. Then it is very hard to see the yellow paths in figure 1B. Either the figure should be enlarged or the yellow color should be avoided ?

Following the reviewers suggestions Figure 1A and 1B have been exchanged and the yellow part of the ray paths have been removed.

Figure 2: I know that this figure is already busy but I would put figure 4B in here because it helps to better understand what is the origin of the postcursors. Otherwise the notion of postcursors is only explain in the text but in a quite vague way while figure 4B is very clear and it precisely illustrates the phenomena. Either it should be a figure by itself or included in figure 2.

This comment is helpful so we decided to take figure 4B out and form an extra figure as new Figure 2 before Figure 3 (since as the reviewer notes, Figure 3 itself is already quite busy) to help illustrate the origins of S_{diff} postcursors (referenced from the main text in line 67).

Figure 3: 3C is quite small and hard to read. I would put a box around each model case with the color of the waveform that is used in 3A and 3B. And maybe it should appear below the waveform in a row instead of a column so that 3C could be enlarged.

We find this comment helpful and updated Figure 4 (previous Figure 3) as the reviewer suggested. We have enlarged Figure 4C, and remade the legend box of waveform right beneath each model. And we think the column layout of the models may be slightly better for comparison. We hope this change will make the model illustration clearer to the audience.

Figure 4: it is a very nice one however 4A is quite large compare to 4B and 4C which are really related to the results of this study. So as I said before I would move 4B into figure 2 and actually enlarge 4C and reduce 4A. For instance 4A could focus on the Hawaiian plume , the rest of the picture is not needed, even not discussed in the text. What is the gray line that appears in the top left corner of 4A for ?

Figure 5 (previous Figure 4) has been adjusted as reviewer suggested. Now have taken the previous 4B out and have made the current figure a left-right layout for A and B, which is now linked into our discussion. The grey line was due to some unexpected figure formatting error, which we have now fixed.

Figure S1: it is strange because on the raw data we clearly see the disappearance of the main Sdiff between 55-65° at long period and the inverse for the postcursors. And this is no longer true after the stacking (Figure 2). Is that a problem with the stacking ? Figure S1 should be after Figures S2 and S3 since they appear first in the text.

In Figure 2 the main Sdiff also shows a much weaker energy. But since we are applying the phase weighted signal, certainly the phase coherence acts as an extra weighting factor in the stacking amplitude, which may increase the main phase energy slightly. To address the reviewer's question, we also performed a simple linear stacking based on the azimuth bins (the result of which is now included as Figure S5 in the supplementary). We see that after stacking there is still some weak energy at the main Sdiff arrival. Our phase weighted stacking may exaggerate the amplitude of coherent energy but the general pattern remains the same. The decaying of the main Sdiff amplitude from 55° to 65° azimuth is indeed interesting and not precisely modelled in this work, as we consider it beyond the scope of this study. Note the work of To et al. 2016 looks at these amplitude reductions and links them to potential defocusing of energy due to large-scale LLSVP structures.

Figure S5. Linear stack of the waveform based on azimuth bins. Linear stacked data of Event 20100320 at S_{diff} distance (100° - 110°) plotted as a function of azimuth, with time axis relative to S_{diff} arrival time predicted by the PREM model. (A) Filtered into long-period range

(10s - 20s) and (B) short-period range (3s - 6s). Yellow colored bands highlight the S_{diff} postcursor.

Figure S3B: the data are very noisy and honestly I would not be able to pick any S_{diff} or postcursors. In general from figures S2 and S3 I am not sure I understand on what arguments is based the drawing of the yellow bands. Can the author give more details in the caption on how they obtain the yellow bands.

We are not sure which event the reviewer particularly refers to within Figure S3B. Indeed the data quality is very variable, and for some events it would be difficult to make exact travel time measurements. We hope the yellow bands, and the synthetic waveforms with postcursors next to the data, help guide the readers eye to the presence of postcursors. We have replotted the yellow bands to they don't extend beyond where we can observe the postcursor as before. We have added a note in the updated caption that the yellow bands are simply sketched in by eye. These features also become easier to identify when directly compared to synthetic data which includes no ULVZ (and thus no post-cursors), but really they are simply more recognisable when one has experience of looking at many of these types of waveforms and knows what to look for!

Figure S4: In the Vs kernel panel, the x-axis should go from 0 to 1.

Figure S4 remade with x-axis from 0 to 1.

Typos:

line 179: bridgmenite – > Bridgmanite

Fixed

line 391-392: "The previous modeled location of show Hawaiian ULVZ sits just southwest of the surface hotspot, centered roughly between 172.5W and 162.5W" – > something is wrong in this sentence.

Equation after line 524 is not well displayed

Wording and equation corrected as reviewer points out.

References

Hauser, J., Sambridge, M., & Rawlinson, N. (2008). Multiarrival wavefront tracking and its applications. *Geochemistry, Geophysics, Geosystems*, 9(11).

To, Akiko, Yann Capdeville, and Barbara Romanowicz. "Anomalously low amplitude of S waves produced by the 3D structures in the lower mantle." *Physics of the Earth and Planetary Interiors*, 256 (2016): 26-36.

Reviewer #2:

Li et al. present a study of the mega-ULVZ beneath Hawaii using S_{diff} waveforms and perform synthetic modeling to refine the previously-published model of the structure (Cottaar and Romanowicz, 2012). The authors update the location of the ULVZ by utilizing data acquired by the transportable array in Alaska which allowed them to expand their analysis

with improved data coverage. The authors also identify frequency-dependent travel times of the Sdiff postcursors and discuss possible small-scale structures associated with the Hawaiian mega-ULVZ.

I find this manuscript interesting and a potentially important contribution to our understanding of the mega-ULVZ beneath Hawaii. The novelty of the work is (1) identification of the high frequency Sdiff postcursors, (2) an update on the location and (3) physical properties of the previous model. I also find that the authors have fully addressed the comments and concerns raised in a previous round by the editor and two reviewers. In my opinion, the manuscript is well-written and all of the figures are publication-ready. I have a few minor points worth further clarification before the manuscript is suitable for publication.

We thank the second reviewer for their positive views of the work and respond to each of their comments below.

The authors define “Sdiff postcursors” as seismic waves that arrive after the main Sdiff phase which show a hyperbolic delayed moveout as a function of azimuth, as expected in the case of a cylindrical ULVZ model (e.g., L63-65). Then the authors state that observations of Sdiff postcursor are quite rare (e.g., L69-70). I think that the definition of Sdiff postcursors and terminology used in the manuscript is misleading because it requires a specific model / interpretation in mind; delayed signals that arrive after the main Sdiff phase, Sdiff postcursors, need not follow a hyperbolic moveout as a function of azimuth, because they can be generated by ULVZs with different morphologies and by other lower mantle structures. In such cases, the “Sdiff postcursors” would be left out of the analysis in this manuscript because they would not necessarily show a hyperbolic moveout. Sdiff postcursors are in fact pervasive and observed on many paths across the Pacific basin (e.g., Kim et al., 2020). That structures capable of producing Sdiff postcursors are widespread is also found using independent datasets (e.g. Thorne et al., 2021). Only in areas of excellent data coverage, and when the ULVZs are unusually large, such as near Hawaii (Cottaar and Romanowicz, 2012) and Iceland (Yuan and Romanowicz, 2017), is hyperbolic delayed moveout as a function of azimuth unequivocally observed. Therefore, I suggest rephrasing the aforementioned text for clarity.

Yes we agree with the reviewer that Sdiff postcursor energy may be pervasive and that the observation of Sdiff postcursor waves with a hyperbolic move-out is still quite rare. We are quite specific that for the purpose of this paper we refer to directly to postcursor observations with a move-out. However, since the later statement causes confusion, we now repeat the definition here (in **line 71-74**), and note the additional postcursors the reviewer refers to:

“...the observation of Sdiff postcursor waves with a hyperbolic move-out is still quite rare (though evidence of Sdiff postcursors without this characteristic, which may be caused by other lower mantle structures, have been observed across the Pacific (Kim et al., 2021, Thorne et al., 2021)).”

<High-frequency Sdiff postcursors>

In my opinion, one of the key observations presented in this manuscript is the identification of the high-frequency Sdiff postcursors. I agree that these phases can provide strong constraints for determining the fine structure of the mega-ULVZ as discussed in the main text. Therefore, the authors should show more examples of these new observations and document their robustness in the manuscript. Currently the authors only show such high-frequency data from a single earthquake (e.g, L105-106; event 20100320) but also acknowledge that high-frequency postcursors are difficult to observe (e.g., L108-109). Can they include some more data examples that display high-frequency postcursors from other events (similar to Figure S1)? Maybe using the single event was intentional, and postcursors

in other event data are contaminated by other arrivals, e.g., depth phases, but I cannot find any further explanation about the choice of event in high-frequency waveform analysis. I think this verification step is crucial since this is one of the most distinctive elements of the manuscript and an independent new constraint on the detailed structure within the Hawaiian mega-ULVZ.

We acknowledge that only one event (20100320) is used in this manuscript for the analysis of high-frequency postcursors. This was also noted as a limitation by the other reviewer. The high frequency signal is quite rare and observations are also limited to many factors that affect the signal noise ratio, e.g. interference from the depth phases, orientation of the source radiation pattern, and magnitude of the event. Though we explored other events in figure S2 and S3, we were not able to extract any additional convincing high-frequency postcursors at 3-6s from the data. We have added some text to that regard **in lines 485- 488**:

“While some of these events have a hint of a high frequency further delayed postcursors, none were of convincing quality to allow interpretation. However, the overall noisy data and presence of depth phases, do not allow us to definitively confirm the absence of high frequency postcursors from this data set.”

We do note that since the first submission of the paper (in November 2020), another PhD student in the group, Carl Martin, has much expanded the search and found many new geometries of data with postcursors as well as high-frequency postcursors for the Hawaii (as well as Iceland) ULVZs. This will be published as a separate first-author paper for Carl.

I understand that the waveforms shown in Figure 2D are highly processed in order to bring out coherent signals between 40-80s. How do the waveforms look in Figure 2D if you were to apply a simple linear stack to Figure S1B? The highlighted band really helps to focus what the authors want me to see, but without the band I would also argue that there are very weak arrivals between 20-40s similar to the long-period postcursors in Figure S1. Am I dreaming here?

The reviewer would like to see a simple linear stack to avoid any possible biases in our processing. We understand this point and thus performed a simple linear stack of the waveforms for each azimuth bins as the reviewer requests. This linear stacking does not assume any prior information except the PREM predicted Sdiff waves along their great circle paths. We see that this linear stack does improve the signal-noise ratio slightly but still shows weak energy arrivals. We have added this linear stack as a new figure in the supplementary (Figure S5) to consolidate our observations and hope this result will address the review’s concern. It illustrates nicely the importance of the directional stacking we apply in the rest of our analysis.

While there could be some energy between 20-40s in the high frequency data, it is not as clear and coherent than the 50-80s postcursor analysed in this manuscript, and is very difficult to pick out in raw data, or linearly stacked data and is thus not further explored in this work.

Figure S4. Original data before stacking. Original data of Event 20100320 at S_{diff} distance (100° - 110°) plotted as a function of azimuth, with time axis relative to S_{diff} arrival time predicted by the PREM model. (A) Filtered into long-period range (10s - 20s) and (B) short-period range (3s - 6s). Yellow colored bands highlight the S_{diff} postcursor.

Figure S5. Linear stack of the waveform based on azimuth bins. Linear stacked data of Event 20100320 at S_{diff} distance (100° - 110°) plotted as a function of azimuth, with time axis relative to S_{diff} arrival time predicted by the PREM model. (A) Filtered into long-period range (10s - 20s) and (B) short-period range (3s - 6s). Yellow colored bands highlight the S_{diff} postcursor.

<Modeling of the ULVZ>

As discussed in the manuscript, the waveform modeling approach has tradeoffs across different waveform parameters. The authors mainly focus on three parameters, location / radius / velocity variation, but is there a reason not to include height in the analysis? As shown and discussed in the supplement (e.g., Figure S2-3; L408-410), the amplitude of the data vs. synthetic waveforms does not show a good fit. The goodness of the fit in this manuscript seems to focus on the backazimuth and delay time of the S_{diff} postcursors but the synthetic waveforms generally underpredict the anomaly. Do you think this can be due to a sub-optimal height of your anomaly in the modeling? I doubt that all of the observed

amplitude discrepancies can be explained by inaccuracies in the CMT solutions and the radiation pattern (e.g., L409-410). Can this be partially caused by physical properties of the ULVZ that are being underpredicted by your proposed model?

We note that the amplitude discrepancies referred to previous lines L409-410 are mainly between main phases and depth phases, which are likely to be source effects. We have clarified this in the text current lines 481-482.

“The discrepancies between main phases and depth phases are likely due to inaccuracies in the CMT sources⁴⁰ and the S_{diff} radiation patterns.”

The main constraint of the height we think is from the frequency content of the S_{diff} waves. The trapped energy of certain frequency ranges is sensitive to structure at certain heights as shown in the kernel (Fig. S4). The higher frequency data we observe are sensitive to the basal layer. But for the low frequency i.e. 10-20s waveform, the amplitude discrepancies in the amplitude can be influenced by many factors if not the inaccuracies in the CMT solutions, e.g. the heterogeneity and scatters above the ULVZ, the exact morphology of the ULVZ, and also possibly the core-mantle boundary topography. We didn't manage to fit the amplitude observations with a higher anomaly in our preliminary testing, which would create a stronger postcursor instead of a weakening of the main phase. Including the height as a new parameter dimension in our modelling will also make the parameter space grow quickly. So we end up with the model based on the optimal choice of height based on frequency sensitivity in the previous literature (Cottaar et al, *EPSL*, 2012).

We do note the paper by To et al., *PEPI*, 2016 that model these amplitude effects with broader LLSVP structures (although it is not clear to what extent this would explain the amplitudes in the data towards the Alaska TA). We agree there are still unsolved questions in the amplitudes of the main phase, which we think is slightly beyond the scope of this study, but interesting to explore later.

Despite these comments/questions and reservations, I am excited that the authors are updating their previous models with more data in different frequencies. I hope the authors find my comments constructive and helpful.

They were!

References

To, Akiko, Yann Capdeville, and Barbara Romanowicz. "Anomalously low amplitude of S waves produced by the 3D structures in the lower mantle." *Physics of the Earth and Planetary Interiors* 256 (2016): 26-36.

Cottaar, S., & Romanowicz, B. (2012). An unusually large ULVZ at the base of the mantle near Hawaii. *Earth and Planetary Science Letters*, 355, 213-222.

REVIEWERS' COMMENTS

Reviewer #1 (Remarks to the Author):

Dear authors,

Thank you for your revised manuscript which is in my opinion ready for publication. Thank you for your answers and for having taken into account all my comments.

I have only two tiny remarks:

- line 710: I would avoid using the term "created" and reword the sentence like that "The mesh is GENERATED for PREM model at a 3s period".

- line 761: "Yellow colored bands are sketched by eye highlight the Sdiff postcursor. " Something is wrong with the sentence ? May be you mean "Yellow colored bands are sketched by eye AND highlight the Sdiff postcursor. "

Best,

Reviewer #2 (Remarks to the Author):

Thank you for addressing all of my review comments. Great work!

Response to Reviewers

Reviewer comments are shown in black, with our responses in red. Where responses have led to changes in the manuscript these are **highlighted in bold red font**, and the section, line or figure number in the updated manuscript is referenced.

Reviewer #1:

Thank you for your revised manuscript which is in my opinion ready for publication. Thank you for your answers and for having taken into account all my comments.

We appreciate the first reviewer's kind and helpful comments. We respond to each of their comments below.

I have only two tiny remarks:

- line 710: I would avoid using the term "created" and reword the sentence like that "The mesh is GENERATED for PREM model at a 3s period".

The sentence now corrected to "The mesh is generated for PREM model at a 3s period." at line 451.

- line 761: "Yellow colored bands are sketched by eye highlight the Sdiff postcursor. " Something is wrong with the sentence ? May be you mean "Yellow colored bands are sketched by eye AND highlight the Sdiff postcursor. "

Sentence grammar now corrected.

Reviewer #2:

Thank you for addressing all of my review comments. Great work!

We appreciate the second reviewer's helpful and encouraging comments.